# Capacity, Collision Avoidance and Shopping Rate under a Social Distancing Regime

**DOI:** 10.3390/e25121668

**Published:** 2023-12-17

**Authors:** Haitian Zhong, David Sankoff

**Affiliations:** Department of Mathematics and Statistics, University of Ottawa, Ottawa, ON K1N 6N5, Canada; hzhon012@uottawa.ca

**Keywords:** social distance, shopping rate, Little’s law

## Abstract

Capacity restrictions in stores, maintained by mechanisms like spacing customer intake, became familiar features of retailing in the time of the pandemic. Shopping rates in a crowded store under a social distancing regime are prone to considerable slowdown. Inspired by the random particle collision concepts of statistical mechanics, we introduce a dynamical model of the evolution of the shopping rate as a function of a given customer intake rate. The slowdown of each individual customer is incorporated as an additive term to the baseline value of the shopping time, proportionally to the number of other customers in the store. We determine analytically and via simulation the trajectory of the model as it approaches a Little’s law equilibrium and identify the point beyond which equilibrium cannot be achieved. By relating the customer shopping rate to the slowdown compared with the baseline, we can calculate the optimal intake rate leading to maximum equilibrium spending. This turns out to be the maximum rate compatible with equilibrium. The slowdown due to the largest possible number of shoppers is more than compensated for by the increased volume of shopping. This macroscopic model is validated by simulation experiments in which avoidance interactions between pairs of shoppers are responsible for shopping delays.

## 1. Introduction

Economic recovery during the COVID-19 pandemic led to the imposition of safety measures such as social distancing in indoor gathering places such as stores. Widely applied criteria included capacity limits, such as a percent reduction of the normal zoning or fire regulation capacities, such as 30% or 50% of the normal amount. Other rules consisted of density restrictions on the number of persons per unit area, such as one person per 5 or 10 square metres. While there is sensible safety-based reasoning behind these limits, and much analytical and simulation work on relating store capacity (e.g., Chang et al. [1], Ntounis et al. [2]) or customer density within a store (e.g., Echeverría-Huarte et al. [3], Harweg, Bachmann and Weichert [4], Mayr and Köster [5]) to physical distance restrictions has been performed, there is little theoretical work addressing the detailed relationship between capacity or density restrictions and their financial impact on the enterprise. It is implicitly assumed that restricting the number of customers will simply have a proportional effect on the total amount of shopping that is done. We will argue, however, that this cannot be universally true.

In queuing theory and operations research, a rule, “Little’s law”, which is claimed to hold universally, states that the average number of customers in a stationary system is equal to the intake rate times the average time a customer spends in the system—Little [6]. The key term here is “stationary”. This rule does not apply directly to the startup period before stationarity is achieved. Neither does it take into account conditions that preclude stationarity, such as when the system capacity is exceeded. The limited applicability of the rule is widely understood: “The only requirements are that the system be stable and non-preemptive; this rules out transition states such as initial startup or shutdown [which means that the entire system and its components are in stable operation and are not affected by external events]. … because a store in reality generally has a limited amount of space, it can eventually become unstable… if the arrival rate is much greater than the exit rate, the store will eventually start to overflow.”—Wikipedia [7].

The effect of intake rate, the flow of customers into the store, on total shopping rate is the phenomenon that we study in this paper. We approach it in two very different ways. First, we introduce a simple dynamical model of the evolution of the shopping rate as a function of a given customer intake rate, starting with an empty store. The slowdown of each individual customer is incorporated as an additive term to a baseline value of the shopping time, proportionally to the number of other customers in the store. We explore alternate ways of making this basic model computationally tractable. This is a minimal model linking “microscopic" behaviour to macroscopic quantities, reminiscent of statistical mechanics, with no variability in customer behaviour and no queuing issues. The only variables are the intake rate, the baseline shopping time and a single proportionality coefficient. This coefficient incorporates, without specifying any details, the store area and layout, social distancing regime and customer interaction (collision avoidance) effects. We analytically determine the trajectory of the model as it approaches a Little’s law equilibrium and identify the point of phase change, where equilibrium cannot be achieved. By relating the customer shopping rate to the slowdown compared with the baseline, we can calculate the optimal intake rate leading to maximum equilibrium spending. We determine whether this optimal rate is inside the domain of a Little’s law equilibrium or is on the border of the non-equilibrium behaviour of the model.

For the second approach, we add an actual store layout to the model as well as a stochastic effect attached to individual customers. We also include a set of randomized strategies, common to all customers, for avoiding infringements of physical distance requirements. It is this that determines the link between the microscopic and macroscopic aspects of the simulation, which was assumed in the purely analytical models.

As with our deterministic model, the only control exercised by the store is the timing of customer entrances. Increasing the entrance rate will increase the number of customers in the store, but it will also slow each customer down as they avoid breaching the social distance criterion. Given these countervailing effects, the enterprise wants to maximize total shopping volume.

In a simulation, each shopper “chooses” a random assortment of locations in the store and tries to complete her shopping efficiently while avoiding other customers. Each potential encounter requires a diversion or backtracking of one or both shoppers, slowing them down. This slowdown gets worse as the number of shoppers in the store increases, leading to instances of local gridlock (also termed jamming), until a major proportion of the shoppers become frozen in place, unable to move without infringing on the space of other shoppers. The entrance rate where this occurs is analogous to the highest rate consistent with a Little’s law equilibrium in our macroscopic model.

There is extensive scientific literature on pedestrian dynamics and simulation (e.g., Muramatsun and Nagatani [8], Kouskoulisa, Spyropouloua and Antoniou [9], Nagatani, Ichinose and Tainaka [10]), including studies by simulation of the effect of increasing the number of individuals moving in one direction, opposing directions or two intersecting directions until the point of “traffic jamming”. This extends to open source: Kleinmeier et al. [11], Lopez et al. [12] and commercial (e.g., PedSim, AnyLogic, Simwell, Oasys, Bentley Systems, PTV Group.) applications of pedestrian simulation. None of this work, however, is directly relevant to our project for two main reasons. First, our focus, embodied in our objective function, namely the total volume of shopping during opening hours, does not seem to have a counterpart anywhere in the pedestrian simulation literature and software, where the interest is basically in the smooth movement of large numbers of people. Second, in the crowd movement literature, the movement is either along corridors or walkways from point-to-point or the efficient entrance and exit of a large number of people through bottlenecks or from an enclosed space, not the independent, multi-directional movement of shoppers, each of whom arrives with, or evolves, a complex, unpredictable, but coherent trajectory in our study.

## 2. The Model and Phase Change

Our macroscopic model is deterministic, involving the controlled rate of flow *f* of customers through the enterprise and the fixed physical characteristics of the floor plan expressed by the effective area parameter *c*. Our only assumption is embodied in a simple equation expressing the average delay in shopping time, caused either by collision or, in our current application, by social distancing.

### 2.1. The Model

Let f=1/Δ be the flow rate, i.e., the number of shoppers entering per unit time (in minutes${}^{-1}$). This is the only quantity controlled by the store manager.

Let *L* be the length of shopping list (number of items) and *P* the average price per item, in $, so that M=LP is the total expenditure by each customer.

We require a coefficient *c*, the “effective area” reflecting not only the store area but also other aspects of capacity, such as its configuration in terms of aisles, doorways, display and advertising obstructions, and especially in subsuming the slowdown effect of encounters invoking social distance rules.

Let the variable *n* be the number of shoppers in store at any particular time and the variable *A* to refer to the store time, in minutes, for a customer. We fix $A_{1}$ to be the store time for a customer in an otherwise empty store, i.e., with no interactions with other customers.

Our model rests on a single “quadratic” assumption, that the rate of increase of shopping time over $A_{1}$ is due to the rate of encounters between shoppers, which is proportional to n2, where *n* is the number of shoppers in the store.

### 2.2. The Equilibrium Phase

Equilibrium holds if the number of shoppers entering a store is equal to the number exiting. We assume that the number of encounters for a given shopper is proportional to the (constant) number of other shoppers in the store, n−1, times the length of time *A* she is in the store.

**Theorem** **1.**
*Equilibrium holds iff $f\leq \frac{{\left(c+1\right)}^{2}}{4cA_{1}}$.*

*If $f=\frac{{\left(c+1\right)}^{2}}{4cA_{1}}$, i.e., at the boundary of the domain of equilibrium,*

(1)
$$\begin{array}{cc} A & =\frac{2c}{c+1}A_{1} \end{array}$$


(2)
$$\begin{array}{cc} n & =\frac{c+1}{2} \end{array}$$



**Proof.** Because of the equilibrium condition, n=fA, the number of other shoppers entering while a given customer is in the store. This is an instance of Little’s law. By definition, $A_{1}$ is the time it would take a shopper to complete her shopping were there no other shoppers in the store. Then, via assumption written before Theorem 1,
(3)$$\begin{array}{cc} A & =A_{1}+\frac{(n-1)A}{c}\\ \; & =A_{1}+\frac{fA^{2}-A}{c} \end{array}$$
So:
(4)$$\begin{array}{cc} \; & cA=cA_{1}+fA^{2}-A\\ \; & 0=fA^{2}-\left(c+1\right)A+cA_{1}\\ A= & \frac{\left(c+1\right)\pm \sqrt{{\left(c+1\right)}^{2}-4fcA_{1}}}{2f} \end{array}$$
Thus, for equilibrium to hold,
(5)$$\begin{array}{cc} f & \leq \frac{{\left(c+1\right)}^{2}}{4cA_{1}}. \end{array}$$
At equality, we obtain
(6)$$\begin{array}{cc} f & =\frac{{\left(c+1\right)}^{2}}{4cA_{1}}, \end{array}$$
(7)$$\begin{array}{cc} A & =\frac{2c}{c+1}A_{1}, \end{array}$$
and
(8)$$\begin{array}{cc} n & =\frac{c+1}{2}. \end{array}$$ □

Let *e* = expenditure rate per customer = M/A and E=en= Total expenditure rate per unit time.

To maximize *E* with respect to Δ, given *M* and *c*,
(9)E=nM/A(10)=fM

The equilibrium state that maximizes *E* is then one where Equations (Equation 6) and (Equation 7) hold. However long the average shopper would take in an empty store, almost doubling this time would maximize expenditure rate under equilibrium conditions. I.e., since, realistically, c,n≫1, maximum *E* is achieved when
(11)$$f\approx \frac{c}{4A_{1}}\approx \frac{n}{2A_{1}},\;A=2A_{1}.$$

In the next two sections, we explore how this maximizing equilibrium, as well other equilibria that do not maximize *E*, are approached as customers enter the store (initially empty), one by one.

The fundamental assumption in this work is that the number of interactions between shoppers is proportional to the square of *n*, the number of shoppers, does not lead to analytical expressions for the number of shoppers in the store at any point of time, the time taken for shoppers to complete their visit to the store, or other quantities of interest. These may be calculated empirically for any particular setting of the parameters $M,f,A_{1}$ and *c*, but here we seek more general mathematical expressions for these quantities.

The difficulties arise from the fact that the number of customers, and, hence, the rate of interactions, are step functions, with the steps at integer times. While this is tractable as long as no customers are finished shopping, as soon as the first few customers finish, this will generally occur at non-integer times. Then, we start accumulating nested expressions containing “ceiling” and “floor” functions, having no simple form.

Our first approach in Section 3 is to allow the number of interactions to be proportional to $t^{2}$, where i<t<i+1. This modification leads to a general solution to the problem, although this is expressed partly through recurrences.

The second approach is to retain the stepwise increase in the number of interactions but to round the customers’ shopping times to the nearest integer. This also leads to a general solution, although the limiting behaviour is oversimplified.

## 3. A Continuous Model

Starting with an empty store, we number the clients i=1,2,… according to the time (i−1)/f they enter, which, in our deterministic model, is the same order in which they leave the store. The number of clients in the store at the time individual *j* enters is $\left[j-K_{j}\right]$, where $K_{j}$ is the last individual to leave the store before *j* enters. Prior to the time $J_{1}$ when the first shopper completes her shopping, $j-K_{j}=j$.

The spending rate for any individual in the store in the instant after individual *j* enters is $\frac{M}{A_{1}+\left[j-1-K_{j}\right]/c}$, so the total spending rate is $\left[j-K_{j}\right]\frac{M}{A_{1}+\left[j-1-K_{j}\right]/c}$.

For tractability, we introduce the approximation
(12)$$\int_{x=0}^{J_{1}-1}\frac{1}{1+\frac{x}{cA_{1}}}dx=Mf,$$
in which the shopping rate decreases slightly in the interval between the entry of the j−1-st and *j*-th shopper, instead of remaining constant. Then, we immediately derive
(13)$$Mf=cA_{1}\log\left(1+\frac{J_{1}-1}{cA_{1}}\right),$$

Solving this,
(14)$$\begin{array}{cc} J_{1} & =cA_{1}\left(e^{\frac{Mf}{cA_{1}}}-1\right)+1 \end{array}$$

While the first customer shops during the $J_{1}-1$ intervals, the second customer enters when x=2 and exits when $x=J_{2}$, shopping during the $J_{2}-2$ intervals. Continuing with our approximate model,
(15)Mf=∫x=1J1−111+xcA1dx+∫y=J1−2J2−211+ycA1dy(16)=cA1log(1+J1−1cA1)−log(1+1cA1)+log(1+J2−2cA1)−log(1+J1−2cA1)(17)=cA1log(1+J1−1cA1)(1+J2−2cA1)(1+1cA1)(1+J1−2cA1).


From (Equation 13), we then have
(18)$$\begin{array}{cc} 1+\frac{J_{2}-2}{cA_{1}} & =\left(1+\frac{1}{cA_{1}}\right)\left(1+\frac{J_{1}-2}{cA_{1}}\right) \end{array}$$
(19)$$\begin{array}{cc} J_{2} & =J_{1}+\frac{cA_{1}+J_{1}-2}{cA_{1}}. \end{array}$$

The third customer enters when x=3 and exits when $x=J_{3}$, shopping during the $J_{3}-3$ intervals.
(20)Mf=∫x=2J1−111+xcA1dx+∫y=J1−2J2−211+ycA1dy+∫z=J2−3J3−311+zcA1dz(21)=cA1log(1+J1−1cA1)(1+J2−2cA1)(1+J3−3cA1)(1+2cA1)(1+J1−2cA1)(1+J2−3cA1)

We know from Equations (Equation 13) and (Equation 18) that
(22)$$\begin{array}{cc} \frac{1+\frac{J_{2}-2}{cA_{1}}}{1+\frac{J_{1}-2}{cA_{1}}} & =1+\frac{1}{cA_{1}}, \end{array}$$
so that
(23)$$\begin{array}{cc} \frac{\left(1+\frac{1}{cA_{1}}\right)\left(1+\frac{J_{3}-3}{cA_{1}}\right)}{\left(1+\frac{2}{cA_{1}}\right)\left(1+\frac{J_{2}-3}{cA_{1}}\right)} & =1. \end{array}$$
(24)$$\begin{array}{cc} \frac{cA_{1}-3+J_{2}}{cA_{1}+1}+J_{2} & =J_{3} \end{array}$$

Continuing in this way, we arrive at the recurrence:

**Theorem** **2.**
*For $r\leq J_{1}-1$,*

(25)
$$\begin{array}{cc} \frac{cA_{1}-r+J_{r-1}}{cA_{1}+r-2}+J_{r-1} & =J_{r} \end{array}$$



In general, a customer may leave the store at a non-integer time $J^{*}$, where $r-1<J^{*}<r$, for some r≥1. The shopping rate during the time interval (r−1,r) changes discontinuously at this point. To account for this, we extend the recurrence in Theorem 2 as follows.

**Theorem** **3.**
*For any r>1, suppose $r-1<J^{*}<r$. Then,*

(26)
$$\begin{array}{c} \frac{1}{cA_{1}+J^{*}-2-K_{r-1}}=\frac{J_{r}-J_{r-1}}{cA_{1}+J_{r-1}-r} \end{array}$$


*For all r>1, if there is no $J^{*}$ satisfying $r-1<J^{*}<r$, then*

(27)
$$\begin{array}{cc} \frac{1}{cA_{1}+r-2-K_{r-1}} & =\frac{J_{r}-J_{r-1}}{cA_{1}+J_{r-1}-r} \end{array}$$



**Proof.** Because the total shopping expenditure *M* for all customers is the same, the spending for customers r−1 in the schema below and *r* is the same during the time period when both of them are in the store (B in the schema below, so the shopping expenditure for customer r−1 from time r−1 to time *r* (A + B) equals the shopping expenditure for customer *r* from time $J_{r-1}$ to $J_{r}$ (B + C). So, the spending in A equals the spending in C.


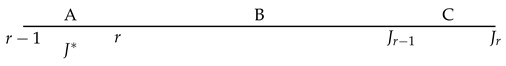


There are two cases. For all r>1, if there exists a $J^{*}$ satisfying $r-1<J^{*}<r$, then:
$$\begin{array}{cc} \int_{x=r-2-K_{r-1}}^{J^{*}-1-K_{r-1}}\frac{1}{1+\frac{x}{cA_{1}}}dx+\int_{x=J^{*}-2-K_{r-1}}^{r-2-K_{r-1}}\frac{1}{1+\frac{x}{cA_{1}}}dx & =\int_{y=J_{r-1}-r}^{J_{r}-r}\frac{1}{1+\frac{y}{cA_{1}}}dy\\ cA_{1}[\log\left(\frac{1+\frac{J^{*}-1-K_{r-1}}{cA_{1}}}{1+\frac{r-2-K_{r-1}}{cA_{1}}})\right]+cA_{1}[\log\left(\frac{1+\frac{r-2-K_{r-1}}{cA_{1}}}{1+\frac{J^{*}-2-K_{r-1}}{cA_{1}}})\right] & =cA_{1}[\log\left(\frac{1+\frac{J_{r}-r}{cA_{1}}}{1+\frac{J_{r-1}-r}{cA_{1}}})\right]\\ \frac{1}{cA_{1}+J^{*}-2-K_{r-1}}=\frac{J_{r}-J_{r-1}}{cA_{1}+J_{r-1}-r} & \end{array}$$In the second case, for all r>1, if there is no $J^{*}$ satisfying $r-1<J^{*}<r$, then:
$$\begin{array}{cc} \int_{x=r-2-K_{r-1}}^{r-1-K_{r-1}}\frac{1}{1+\frac{x}{cA_{1}}}dx & =\int_{y=J_{r-1}-r}^{J_{r}-r}\frac{1}{1+\frac{y}{cA_{1}}}dy \end{array}$$
(28)$$\begin{array}{cc} cA_{1}[\log\left(\frac{1+\frac{r-1-K_{r-1}}{cA_{1}}}{1+\frac{r-2-K_{r-1}}{cA_{1}}})\right] & =cA_{1}[\log\left(\frac{1+\frac{J_{r}-r}{cA_{1}}}{1+\frac{J_{r-1}-r}{cA_{1}}})\right] \end{array}$$
(29)$$\begin{array}{cc} \frac{1}{cA_{1}+r-2-K_{r-1}} & =\frac{J_{r}-J_{r-1}}{cA_{1}+J_{r-1}-r} \end{array}$$
□

Theorems 2 and 3 permit us to calculate an exact description of the process given any values of the store and shopping parameters. These are not closed forms, but they allow for the rapid and precise calculation of the number of shoppers at all times.

We can graphically display the behaviour of $J_{r}$ and related quantities using a specific example.

The parameter values should satisfy the following:(i)The parameter $A_{1}$ represents the total shopping time for a customer in an empty store, i.e., one with no other customers. So, Δ should be less than $A_{1}$, and considerably so, for the model to be interesting. And f=1/Δ represents the number of customers who enter the store per unit of time. Thus,
(30)$$\begin{array}{c} A_{1}*f>1 \end{array}$$(ii)Because there will be a continual stream of customers entering the store, the shopping time for the first customer, $\frac{J_{1}-1}{f}$, should be greater than $A_{1}$, so
(31)$$\begin{array}{cc} J_{1}-1 & =cA_{1}\left(e^{\frac{Mf}{cA_{1}}}-1\right)>A_{1}*f\\ ⟹M & >\frac{cA_{1}}{f}*\ln\left(\frac{f}{c}+1\right) \end{array}$$

One realistic scenario is where $M=0.137,c=70.58,A_{1}=0.2108$ (in hours). Changing the parameter values will change the scale of the results but not their basic form.

Figure 1 shows $J_{r}$, the total number of customers entering before individual *r* leaves the store. Then, $\Delta (J_{r}-r)$ is the total shopping time for the *r*th customer. In this figure, for each Δ, later customers, i.e., larger values of *r*, use increasing amounts of time to complete their shopping because more customers are in the store, slowing down each customer’s shopping rate. By the same token, as Δ decreases, the store fills up at a faster rate and the entire trajectory of $J_{r}$ takes on higher values.

Figure 2 represents the number of customers in the store during the first 20,000 s for various values of Δ. In this figure, we find that each curve for each Δ≥42 s asymptotically approaches a constant. For Δ≤41 s, on the other hand, $J_{r}$ increases without bound.

The curves for all Δ<42 s tend to infinity. In this range, then, our model becomes increasingly unrealistic, because it suggests the store will become impossibly crowded. But in reality, and in our simulations in Section 5 below, low values of Δ are reflected in the shoppers’ being jammed up or frozen in place in some part of the store.

In Figure 3, we can see the number of customers who have completed shopping during 5000 s. Although we did not define a criterion for “freezing” in our mathematical model, we can draw some conclusions about it from this figure. In our model, customers will complete their shopping and leave the store one by one in the order that they entered the store. So, up to a point, with a higher flow rate, more customers will complete the shopping during the same time interval. However, it is obvious that this curve reaches its maximum near 42 s. This reflects that when the flow rate is too large, fewer shoppers leave than enter because of frequent interactions, and the number of shoppers increases without limit. Whatever the size of the store, this will sooner or later result in extreme congestion, which, in our simulations in Section 5 below, leads to freezing. When the flow rate is not too large, the store can run stably and shopper numbers will tend to equilibrium.

Figure 4 shows that the first shopper exits at a time only sightly larger than $A_{1}$ when the entry interval is 60 s but is slowed down somewhat at 42 s and takes about 50% longer when Δ=30 s.

In Figure 5, we see that the shopping time $A=\frac{2c}{c+1}A_{1}=1496.36$ s is achieved at Δ=41.8 s as predicted by the theory in Section 2.

Looking at the rates for the Δ≥42 s curves in Figure 6 we see they approach a Little’s law equilibrium. The higher, and unbounded, values associated with Δ<42 pertain to the unrealistic situation where the number of customers is not bounded. In other words, our model allows us to set a lower bound on Δ, beyond which the operation of the store is not sustainable.

The overall conclusion is that the way to optimize total spending while keeping the number of shoppers stable is to allow as many shoppers to enter as possible, getting as close as possible to the bifurcation point without exceeding it.

## 4. The Dynamics of Store Time

In this section, we explore an alternative formulation of the same problem discussed in Section 3, where the number of individuals in the store increases as a step function instead of having continuous growth. The assumption that the $J_{r}$ are all integer values is closer to reality than that of the non-integer values in the continuous model. On the other hand, it involves rounding errors in the exit times accumulating to produce errors once the first shopper leaves the store.

### 4.1. The Initial Trajectory

The spending rate for the first individual after the *j*th individual enters is $\frac{M}{A_{1}+\left[j-1\right]/c}$ so that she buys at only $\frac{A_{1}}{A_{1}+\frac{j-1}{c}}$ of her best rate. Multiplying this rate by the length of time interval $\frac{1}{f}$ will give the expenditure during this time interval. Then, she will have to wait until the $J_{1}$-st individual enters to finish spending *M*, where
(32)$$\begin{array}{cc} M & =\frac{1}{f}\sum_{j=1}^{J_{1}-1}\frac{A_{1}}{A_{1}+\frac{j-1}{c}}\\ Mf & =\sum_{j=1}^{J_{1}-1}\frac{A_{1}}{A_{1}+\frac{j-1}{c}}\\ \; & =\sum_{j=1}^{J_{1}-1}\frac{1}{1+\frac{j-1}{cA_{1}}}\\ \; & =(\sum_{k=1}^{J_{1}-2}\frac{1}{1+\frac{k}{cA_{1}}})+1, \end{array}$$
under the assumption that $J_{1}$ is an integer. The effects of this assumption, which extends to the shopping completion times $J_{2},\ldots$ of subsequent customers, are minimal as long as $A_{1}$ is large with respect to Δ but they nonetheless motivate the continuous version of the model in Section 3. From Equation (Equation 32), we have
(33)$$\begin{array}{cc} \frac{Mf}{cA_{1}} & =\psi \left(cA_{1}+J_{1}-1\right)-\psi \left(CA_{1}\right), \end{array}$$
where ψ is the digamma function. As *x* gets large, ψ(x) approaches logx so that Mf is approximated by
(34)$$cA_{1}\log\left(1+\frac{J_{1}-1}{cA_{1}}\right).$$
which is identical to the form of Equation (Equation 13) we derived in the continuous situation, representing the limiting behaviour of the discrete model. But we have found that this assumption represents a very slight departure from reality.

We now study the case of shoppers who enter the store before the first shopper leaves, namely shoppers $r=2,\ldots ,J_{1}-1$.

Examining when the second shopper leaves the store, i.e., when the $J_{2}$-nd shopper enters, we have
(35)$$\begin{array}{cc} Mf & =\sum_{j=2}^{J_{1}-1}\frac{1}{1+\frac{j-1}{cA_{1}}}+\sum_{j=J_{1}}^{J_{2}-1}\frac{1}{1+\frac{j-2}{cA_{1}}}\\ \; & =\sum_{k=1}^{J_{1}-2}\frac{1}{1+\frac{k}{cA_{1}}}+\sum_{k=J_{1}-2}^{J_{2}-3}\frac{1}{1+\frac{k}{cA_{1}}}\\ \; & =\sum_{k=1}^{J_{2}-3}\frac{1}{1+\frac{k}{cA_{1}}}+\frac{1}{1+\frac{J_{1}-2}{cA_{1}}} \end{array}$$

From Equations (Equation 32) and (Equation 35),
(36)$$\begin{array}{cc} \sum_{k=1}^{J_{1}-2}\frac{1}{1+\frac{k}{cA_{1}}}+1 & =\sum_{k=1}^{J_{2}-3}\frac{1}{1+\frac{k}{cA_{1}}}+\frac{1}{1+\frac{J_{1}-2}{cA_{1}}}\\ 1 & =\sum_{k=J_{1}-1}^{J_{2}-3}\frac{1}{1+\frac{k}{cA_{1}}}+\frac{1}{1+\frac{J_{1}-2}{cA_{1}}}\\ \; & =\sum_{k=J_{1}-2}^{J_{2}-3}\frac{1}{1+\frac{k}{cA_{1}}} \end{array}$$

Examining when the third shopper leaves the store, i.e., when the $J_{3}$-rd shopper enters, we have
(37)$$\begin{array}{cc} Mf & =\sum_{j=3}^{J_{1}-1}\frac{1}{1+\frac{j-1}{cA_{1}}}+\sum_{j=J_{1}}^{J_{2}-1}\frac{1}{1+\frac{j-2}{cA_{1}}}+\sum_{j=J_{2}}^{J_{3}-1}\frac{1}{1+\frac{j-3}{cA_{1}}}\\ \; & =\sum_{k=2}^{J_{1}-2}\frac{1}{1+\frac{k}{cA_{1}}}+\sum_{k=J_{1}-2}^{J_{2}-3}\frac{1}{1+\frac{k}{cA_{1}}}+\sum_{k=J_{2}-3}^{J_{3}-4}\frac{1}{1+\frac{k}{cA_{1}}}\\ \; & =\sum_{k=2}^{J_{3}-4}\frac{1}{1+\frac{k}{cA_{1}}}+\frac{1}{1+\frac{J_{1}-2}{cA_{1}}}+\frac{1}{1+\frac{J_{2}-3}{cA_{1}}} \end{array}$$

From Equations (Equation 35) and (Equation 37),
(38)$$\begin{array}{cc} \sum_{k=1}^{J_{2}-3}\frac{1}{1+\frac{k}{cA_{1}}}+\frac{1}{1+\frac{J_{1}-2}{cA_{1}}} & =\sum_{k=2}^{J_{3}-4}\frac{1}{1+\frac{k}{cA_{1}}}+\frac{1}{1+\frac{J_{1}-2}{cA_{1}}}+\frac{1}{1+\frac{J_{2}-3}{cA_{1}}}\\ \frac{1}{1+\frac{1}{cA_{1}}} & =\sum_{J_{2}-3}^{J_{3}-4}\frac{1}{1+\frac{k}{cA_{1}}} \end{array}$$

The calculations above illustrate an induction argument by which we can prove the following:

**Theorem** **4.**
*For $r\leq J_{1}-1$,*

(39)
$$\begin{array}{cc} \frac{1}{1+\frac{r-2}{cA_{1}}} & =\sum_{J_{r-1}-r}^{J_{r}-r-1}\frac{1}{1+\frac{k}{cA_{1}}} \end{array}$$



This is the analog of Theorem 2 in the continuous model.

### 4.2. Approach to Equilibrium

Theorem 4 opens a way to the computation of the $J_{r}$ for the customers who enter before $J_{1}$. The situation is somewhat more difficult for the customers who enter later. We use the same idea used to prove Theorem 3.

We denote with $K_{r}$ the number of customers who have left the store up to and including the moment the *r*-th customer enters. So, if $J_{k^{*}}\leq$r$<J_{k^{*}+1}$, then $K_{r}=k^{*}$.

Recalling our discussion of $K_{r}$ at the beginning of this Section, we rewrite Theorem 4 as

**Theorem** **5.**

(40)
$$\begin{array}{cc} \frac{1}{1+\frac{r-(K_{r-1}+2)}{cA_{1}}} & =\sum_{i=J_{r-1}-r}^{J_{r}-r-1}\frac{1}{1+\frac{i}{cA_{1}}} \end{array}$$



This version of the theorem allows us to compute $J_{r}$ (and $K_{r}$) for all *r*, not only those covered by Theorem 4. The recurrence in the theorem is not, however, a closed form that would allow us to find a value of *r* for which equilibrium is attained.

Nevertheless, we can prove some strong results about $J_{r}$ and $K_{r}$ in the equilibrium phase. Since customers enter once every time interval, one customer has to leave every time interval. Then, $J_{r}-J_{r-1}=1$ if the equilibrium condition is met for r−1 or earlier. With some effort, we can prove the following:

**Theorem** **6.**
*If $J_{r}-J_{r-1}=1$ occurs, then equilibrium is reached from time $J_{r-1}$ and the number of customers in the store will always satisfy $J_{r}-r$.*

*And $J_{r}=2r-\left(K_{r-1}+1\right)$ will always hold starting from some time $r_{0}$ which is later than time $J_{1}$.*


$J_{r}$ will eventually reach $J_{r}^{equilibrium}=2r-\left(K_{r-1}+1\right)$ and remain at that value. If $J_{r}$ does not satisfy this equation, equilibrium has not yet been reached.

To discover the exact condition for equilibrium, we study the difference between the value of $J_{r}$ calculated using Theorem 5 and the hypothetical value of $J_{r}$ where equilibrium was reached at the outset. If this difference is 0 at some point, this means that $K_{r-1}=K_{r-2}+1$ has been satisfied, and $J_{r}-J_{r-1}=1$ will hold thenceforth.

With a considerable amount of work, we can prove the following:

**Theorem** **7.**
*If $g\left(r\right)=J_{r}-J_{r}^{equilibrium}$ starts to decrease at time $r_{0}$, earlier than time $J_{1}$, then it will reach equilibrium at time $J_{r_{0}}$ and the number of customers in the store will remain at $J_{r_{0}}-r_{0}$. On the other hand, if g(r) starts to increase before time $J_{1}$, then it will never reach equilibrium and the number of customers will always increase.*


We can also show the following:

**Theorem** **8.**
*The total shopping rate will keep increasing until the maximum is achieved when equilibrium has been reached, although the shopping rate for individual customers will decrease.*


It is remarkable, as Theorem 6 shows, that, if there is to be equilibrium when customer *r* leaves, this is already pre-ordained by the dynamics of the number of customers ($K_{r-1}=K_{r-2}+1$) in the store immediately before the *r*-th customer even enters.

Our characterization of the equilibrium context in Theorems 4–6 can be identified with the parameters of Section 2.1:(41)$$\begin{array}{cc} A & =\Delta (J_{r}-r) \end{array}$$(42)$$\begin{array}{cc} n & =J_{r}-r=2r-\left(K_{r-1}+1\right)-r=r-\left(K_{r-1}+1\right)=r-K_{r} \end{array}$$

To illustrate the effects of entry frequency on $J_{1}$, we picked particular values for the parameters M,c and $A_{1}$, namely $M=0.148,c=70.58,A_{1}=0.2108$ (in hours). Though we could have picked a wide range of other values without changing the general form of the output variables, these particular numbers are in a range that is both realistic and amenable, in terms of computing time, to the simulations reported in Section 5 and Section 6.

Recall that the *r*th customer enters the store at the beginning of the *r*th time interval. And the value of $J_{r}$ represents the number of time intervals which have passed when *r* leaves. Multiplying *r* or $J_{r}$ by Δ converts the “number of intervals” scale to clock time.

In working out the example, we had to come to terms with the fact that the formal development in this Section requires that the $J_{r}$ are integer-valued, which leads to small errors. For example, the sum in Equation (Equation 32) will exceed Mf, but if we had left off the $J_{1}^{st}$ term, the sum would have been smaller than Mf. My practical solution was to choose whichever sum was closest to Mf. In actual calculations, normalizing the interval between the two sums to [0,1], it turned out that keeping the $J_{1}^{st}$ was preferable only when the sum was within 0.465 (not 0.5) of the desired value, and, otherwise, it was preferable to drop this term.

As with the continuous model, the value of $J_{r}$ will increase when the entry interval decreases. However, we can see a big difference in Figure 7.

This graph reveals a bifurcation in the pattern of the $J_{r}$ curves as Δ changes. The curves for all Δ≥42 s follow a common pattern that is different from the common pattern for Δ≤41 s. The slopes of the curves for the larger values of Δ tend to 1, while the slopes of the curves for the smaller values of Δ tend to 2. A slope greater than 1 means customers are entering faster than earlier customers are leaving. This will result in continuously increasing numbers of customers in the store and the impossibility of equilibrium. This is clear in Figure 8, where the number $L_{r}$ of customers in the store when *r* enters is
(43)$$\begin{array}{c} L_{r}=r-K_{r}. \end{array}$$

In Figure 8, when Δ≥42 s, $L_{r}$ eventually attains a Little’s law equilibrium. But, for Δ≤41 s, $L_{r}$ increases without limit. My model does not require specifying store capacity, but this could easily be incorporated into our calculations to find the point at which the store will overflow, or customers will no longer be able to move around. In Section 5 of this paper, we will show how this “pathology” plays out in a more realistic model.

In Section 2.2, we obtained conditions for equilibrium. From Theorem 1 and the particular model parameters used to illustrate the model, it can be calculated that the condition for equilibrium is
(44)Δ=1/f(45)≥4cA1(c+1)2(46)=41.8s,
which agrees with the bifurcation observed in Figure 7 and Figure 8.

It is instructive to plot the number of customers who complete shopping within a substantial time after the store has been open for, say, 5000 s, as in Figure 9.

When the entry interval Δ is small (≤41 s) or, equivalently, the entry frequency *f* is large, the number of customers who enter the store up to any given time may well be greater than when there is more severe restriction on entry, but the number of customers who succeed in completing their shopping is less when the entry interval is too small, namely 41 s or less in our model.

The number of customers who complete their shopping differs very much between ≤41 s and ≥42 s. This is because, if there is no equilibrium, the rapidly rising number of customers will slow each other down and their spending rate will decrease so quickly that few of them will be able to complete their total shopping list. As for the entry interval which is ≥42 s, customers complete their shopping lists quickly and at a steady rate. Thus, the number of customers who complete their shopping will be greater even though fewer customers will have entered the store.

Figure 9 differs dramatically from its counterpart Figure 3 in Section 3. This can be understood through the more sudden difference between the Δ≥42 and Δ≤41 trajectories in Figure 7 and Figure 8 compared with Figure 1 and Figure 2 in Section 3.

Another interesting plot depicts $J_{1}$ with different entry intervals.

The *x*-axis of Figure 10 is in clock time, measured in seconds. All the $J_{1}$ in this figure are larger than 759 s, which is the same as $A_{1}$. This attests to the reasonableness of the selection of the parameter values.

Note that $J_{1}$ is not the point where equilibrium is reached, but that it represents a slower rate of increase in the shopping time for a large Δ, but not for a small Δ.

The shopping time for the customer after the first customer must be longer than the time for the first one. In the figure, the shopping time for the first customer with a 30-s entry interval is 250 s more than the one with a 60-s entry interval. That is almost $\frac{1}{3}$ of the shopping time for the first customer with a 60 s entry interval. This means that, for the 30-s entry interval, the customer has already been affected by many other customers.

One important thing is that, according to the Theorem 7, when we want to make sure that equilibrium can be reached with a specific δ, we just need to pay attention to the customers who enter the store before $J_{1}*\delta$. If there are any two of these customers like $r_{0}$ and $r_{0}+1$ who leave the store 1 min apart, then equilibrium could be reached at time $J_{r_{0}}$.

The saw-tooth effect visible in this figure is an artifact of our $J_{1}$ rounding, which also affects the next few figures; it has no interest beyond that.

Figure 11 shows a plot of the average shopping time of a customer who has completed shopping within 5000 s. This is parameter *A* in Section 2.

Figure 11 includes a horizontal and a vertical line. These highlight the fact that when Δ≥42 s, the average shopping time of customers who have completed their shopping list will be $<\frac{2*A_{1}*c*3600}{c+1}=1496.36$ s. That the maximum shopping time under equilibrium is $A=\frac{2*A_{1}*c}{c+1}$ can be understood in terms of Theorem 1. On the other hand, if the entry interval is ≤41 s, the average shopping time will be much greater.

Besides that, these data could help to compare the formula of Equation (42) with n=fA used in Theorem 1. The data in Figure 11 are represented by *A*. This gives a comparative picture for entry intervals that are ≥42 s because they approach equilibrium (see Figure 12).

In Figure 12, the black points are from the calculation of the discrete model, and this kind of data is obtained from Figure 8 directly. Then, the red points are obtained from the product of the data plotted in Figure 11 and the corresponding entry interval. These two sets of data are very close. There is a little error between them because of the use of approximation when calculating $J_{r}$. But this does not affect the final conclusion; that is, our formula in Equation (42) and theory in Section 2.1 are consistent.

We conclude this section with the most important result: the total spending rate. This is depicted in Figure 13, which illustrates the sum of the spending rate of each customer currently in the store. Two patterns can be distinguished. For those Δ which are too small to result in an equilibrium flow of customers, without equilibrium, their number will keep increasing unrealistically and there will be no stable spending rate.

The other pattern in Figure 13 includes the total spending rate when Δ is from 42 s to 60 s. These curves will be flat after a specific time and this means they will all have stable spending rates. Moreover, the total spending rate will increase as the entry interval decreases. Although there will be more customers in the store, which slows down the spending rate for each customer, this effect is not large enough to negate the positive effect of *n*, the stable number of the customers. Our total spending rate is equal to the number of customers multiplied by each customer’s spending rate. With this model, as with the previous one, the conclusion is that letting as many customers in the store as is compatible with stability will be the most profitable. The irregular nature of some of the results, however, requiring explanation in terms of the rounding off of exit times, makes this model less attractive than the one in Section 3.

## 5. Design of the Simulations

### 5.1. Overview

We model the freedom of shoppers to formulate their own shopping lists in advance with a random, normally distributed number *L* of items located at random points along the display cases around the store. This placement of the items also models a passive role of management. The customer enters and travels from one of their list items to the closest next one, using a line-of-sight approach or proceeding to the end of the aisle (the closest corner of an internal display) until completion, when they exit. Over the course of their shopping, customers may have to pause or deviate from their trajectory to avoid collisions (i.e., breaches of the social distance criterion). The dispersed distribution of *L* instead of the fixed total cost *M* in the discrete and continuous models is a change that no longer allows for the assumption that customers enter and leave in the same order.

The store was modelled as a rectangular layout with one four-sided display in the centre. There would be little difficulty in incorporating layouts involving any number of displays, counters, walls, aisles (possibly one-way), checkout lines, other waiting lines and even multiple floor levels without substantially changing the objective function, the control parameter (entrance timing) or the simulation principles. In contrast, the discrete and continuous models needed no specific layout or even total area. All that was necessary were the parameters $A_{1}$ and *c*.

Included with the simulation is an animated visualization. Each customer is represented by a coloured dot, and her shopping points are represented by colour-coded ×’s along the boundary, as in Figure 14. The trajectory of the customers and their interactions can be followed in the actual time scale. The appearance and disappearance of shopping points are visible, and a separate display shows an alert each time there is an interaction, with the IDs of the two customers involved.

The main difference between the models in Section 2, Section 3 and Section 4 and the simulation discussed in the present section is that there is no longer the assumption that each of the *n* customers has an equal chance, proportional to $\frac{1}{n-1}$, of an interaction with each other customer, so that interactions happen at a rate proportional to n2. Instead, every customer pursues her own shopping goals in the store, based on an efficient trajectory of moving from one item to another, randomly chosen at the outset. Whenever there is an immediate danger of social distance infringement with another customer, one or both of them take evasive action, which prolongs their shopping time.

Another important difference is how departures from stability are manifested. In the models, this simply shows up as an increase in the number of customers without limit. In the simulation it is seen as a freezing or “jamming” [8,10] situation in some part of the store so that a large number of shoppers are stuck and the simulation cannot be completed, as on the right in Figure 14. Indeed, much of the research on this project was dedicated to avoiding, or at least delaying, the freezing phenomenon, but, given the strict infringement avoidance rules together with the goal of each customer to get to a nearby shopping point, some freezing is inevitable. We believe that more sophisticated behavioural simulations and models of shopper variability will reduce or delay this phenomenon, but not by much.

Figure 14 shows a typical freezing situation on the right. Many customers are stuck around the lower part of the store. Some of them need to go out through exit; note that there are few shopping points left compared with the left panel, but some other customers are still shopping. Since they all have to maintain social distance, they find themselves unable to advance to their goals.

### 5.2. Simulation Details

The setup for the simulation is as follows:We designed a 30 m by 30 m store with a 10 m by 10 m display counter in the centre. The shopping points can occur along all four store walls and all four edges of the display counter.The number of items on the shopping list of each customer follows a normal distribution with a mean of 15 and a variance of 2. These items are spatially distributed uniformly and randomly along the walls and display case.To avoid freezing around the entrance, each shopper’s trajectory is initiated at one of her shopping points, chosen at random. Every customer will go to this shopping point first, then complete the original shopping list as described below. This is implemented primarily to avoid the repeated premature termination of simulations, but it may also be interpreted as the customer’s desire to avoid waiting for her first shopping point if there are too many customers.After each shopping point, including the first one, the customer chooses the nearest line-of-sight shopping point remaining on her unfinished shopping list, if there is one, to be the next shopping point. The customer will follow a straight line between her current location and next shopping point.Customers’ walking speed is set at 0.3 m per s. If the customer approaches a shopping point and the distance between her and the shopping point is smaller than 0.3 m, then her next step is set to be this shopping point. Customers spend 15 s at every shopping point.If none of the remaining items on her list are in the line of sight, as they may be obscured by the display table, the customer walks in a straight line to the next corner of the display table and then proceeds to the nearest visible shopping point.With the motivation for this work being the impact on store management under a social distancing regime during the COVID-19 pandemic, the maintenance of physical distance is of paramount importance. The simulation is designed so that customers cannot get within two meters of each other.To maintain the two-meter social distance, the customers follow a number of rules to avoid infringing on each other’s space. If a step were to move a customer to within two metres of another person, this step is not taken. Instead, the program consults the other person’s projected trajectory in relation to the current customer. According to which of four quadrants that movement is projected to be, the current customer chooses right (R), left (L), back (B) or wait (W) with the given probabilities, as in Figure 15. If an R, L or B move were to cross a wall or a display edge, it is replaced by W. If the customer chooses B, a number *s* of steps backward will be randomly taken with a probability of $1/2^{s}$.At each point in time, the movement of the customers is calculated sequentially, in the order they entered the store.

### 5.3. Reducing the Quadratic Coefficient for Collision Detection

Since the number of customers who can fit in the store is bounded, the computing time requirements for a simulation will be proportional to the duration of the experiment. In the simplest possible conception, after each time step, of which there are thousands, each customer has a position and a direction. Before the next step, the simulation program has to check whether it will bring any two customers into a breach of the physical distancing criterion, an $O(n^{2})$ task. However, even with a moderate number of customers in the store, this is computationally very expensive over the duration of the experiment and over the many repetitions of the experiment necessary to make generalizations. It is necessary to avoid calculating all pairwise distances between them too often, though, for any two customers in the store at the same time, this distance must be calculated at least once. To keep to this minimum, we make use of a “risk time” linked list, which contains the shortest possible time that the distance between two customers can be reduced to, in meters.

Each customer *X* carries two vectors. The first vector ${\mathrm{risktime}}_{X}\left(Y\right)$ contains the collision risk time $t_{X}^{\prime }\left(Y\right)=t_{Y}^{\prime }\left(X\right)$. This number is usually calculated only once, when *X* enters the store (if *Y* is already there).
(47)$$t_{X}^{\prime }\left(Y\right)=\frac{\mathrm{dist}(X,Y)}{2\times \mathrm{speed}}-\mathrm{social}\;\mathrm{distance}.$$

Only when $t_{X}^{\prime }\left(Y\right)=t$, the current time, is dist(X,Y) calculated again, so that either we perform collision avoidance (rare) or update $t_{X}^{\prime }\left(Y\right)$ (more usual). Only $t_{X}^{\prime }\left(Y\right)$ is recalculated, not all the other customers. When *X* leaves the store, all $t_{X}^{\prime }\left(Y\right)$ are set to 1,000,000.

The second vector carried by *X*, called ${\mathrm{nextrisk}}_{X}\left(Y\right)$, contains the name/number *Z* which has the next risk time with *X* immediately after *Y*. Finally, there are two numbers associated with *X*, called r(X) and s(X), which contain the earliest time in $t_{X}^{\prime }\left(Y\right)$ and the *Y* which has this earliest time.

When *X* enters the store, $t_{X}^{\prime }\left(Y\right)$ is calculated for all *Y*. Then, these values are ordered, enabling us to define ${\mathrm{nextrisk}}_{X}\left(Y\right)$ and r(X). Also, for all the other *Y* in the store, we have to find the largest $t_{Y}^{\prime }\left(Z\right)$ less than $t_{X}^{\prime }\left(Y\right)$, where ${\mathrm{nextrisk}}_{Y}\left(Z\right)=W$, and change these to ${\mathrm{nextrisk}}_{Y}\left(Z\right)=X$ and ${\mathrm{nextrisk}}_{Y}\left(X\right)=W$.

Then, *X* starts shopping until time r(x). At that time, we check s(X). If s(X)=Y, we calculate dist(X,Y). If this is greater than the socialdistance, we recalculate $t_{X}^{\prime }\left(Y\right)$ and insert it appropriately into nextrisk(X). Otherwise, we do collision avoidance before recalculating $t_{X}^{\prime }\left(Y\right)$. Then, we change r(X) and s(X) by consulting ${\mathrm{nextrisk}}_{X}\left(Y\right)$. And *X* continues shopping.

Note that any time we change $t_{X}^{\prime }\left(Y\right)$ and adjust nextrisk(X), we have to change $t_{Y}^{\prime }\left(X\right)$ and adjust nextrisk(Y).

Also note that when *X* enters, all of the $t_{X}^{\prime }\left(Y\right)$ must be calculated, and they must be ordered, to construct the linked list nextrisk(X). But once *X* starts shopping, there is no more long calculation. If *X* is far from everybody else, r(t) will be large and there will be many steps with no calculation. All we have to do is occasionally update $t_{X}^{\prime }\left(Y\right)$ and r(t) and maybe nextrisk(X) and s(t), which are just a few steps.

## 6. Results

For the simulations, 16 values of Δ were sampled, from 30 to 65 in five-second intervals, as well as 28 s, 31, 32, 33, 34, 37, 42 and 47. The unequal spacing was designed to span intervals where important changes in behaviour were expected. For each of these values, 100 successful simulation experiments were conducted, with customer positions recalculated every second up to 3600 s. “Successful” means that no freezing took place. All unsuccessful simulations, terminated by freezing events, were discarded and replaced by additional experiments.

To determine the rates of freezing, in Figure 16, the proportion of all experiments, successful and unsuccessful, are plotted. Thus, the 40% successful simulations where Δ=28 s are based on the 100 experiments to be used in subsequent analyses, plus 150 discarded unsuccessful experiments.

Operationally, if a customer stays at one point for more than 200 s, it is considered to be a freezing situation. If a customer enters the store at time *i*, but is unable to move from the entrance at time i+Δ, this was also considered to be freezing.

The practice of discarding unsuccessful experiments is unavoidable. Replacing them until a successful experiment is recorded was a choice made in the name of ensuring the collection of enough data to accurately characterize the model. Both practices undoubtedly introduce biases into estimates, but how severe these are is not clear. We conjecture that this gives results that portray the simulations with small Δ as being more favourable to total spending.

From Figure 16, it is clear that when the entry interval decreases so that more customers enter the store per unit time, the probability of a successful experiment will decrease.

For all the variables we consider, we take the mean over 100 simulations for each of the 3600 s of the simulation.

In Figure 17, the line for Δ=40 s is the grey line in the middle, and the red line below it represents 42 s. A blank area is apparent between these two lines. Each of the curves below this blank area tend toward a constant value, while the other curves display increasing growth. This means that the number of customers in the store is stable after the initial growth when the entry interval is equal to or bigger than 42 s. But for entry intervals which are equal or smaller than 40 s, although 3600 s worth of data can be extracted from these experiments, the growth will eventually succumb to freezing. Already at 28 s, this happens more than half of the time.

Figure 18 reveals how many customers have completed their shopping list from time 0 to time 3600. This curve increases first and decreases later with the maximum value occurring in between. The decrease is understood from Figure 19, simply because there are fewer customers entering the store, with a negligible decrease in shopping time. But why does the curve in Figure 18 increase at first? The reason for this situation can be found in Figure 17. It is because, when more and more customers stay in the store, the probability of collision for each customer increases significantly, so they cannot access the shopping point directly. As a result, customers need more time to complete their shopping lists and fewer customers will complete them before 3600 s.

In addition, the largest value is 50.04, which occurs when Δ is 45 s. This means that an average of 50.04 people completed shopping during 3600 s when the time period was 45 s. However, the numbers of customers who completed shopping are all greater than 49 for all Δ from 40 to 50 s. The effect of these entry intervals is similar.

Figure 20 shows how long customers take to complete their shopping (only those who have completed their shopping before 3600 s). This curve keeps decreasing because when the entry interval is larger, the number of customers entering the store in the same time interval will decrease, and the probability of collision will decrease. In this way, customers arrive at their shopping points and exit without many detours. Their shopping time will approach $A_{1}=758.78$ s.

### Accessibility

The simulation codes in this subsection can be found in the following link: https://github.com/css614/Thesis-codes/blob/main/Thesis-code.R (accessed on 12 August 2021).

These codes run in RStudio 3.6.1. The scale of the store and customers’ walking speed are preset, so the most important parameter in these codes is the “entrance interval”. When this decreases, it will take a longer time to run the whole code and it will be more susceptible to freezing. If an experiment freezes, an error code is generated.

The output generated and stored by this code includes the total number of customers who enter the store, the location of each customer at every second, the shopping list of each customer, the shopping time of each customer, the spending rate of each customer at every second, the times of direction change by each customer and the state of each loop.

## 7. Conclusions

Whether or not our simulation experiments are realistic, given that customers behave identically except for their entrance time and their shopping list, is a subjective question. Certainly, the animations give the impression of normal activity—even to the extent that the observer is tempted to impute strategies, personality traits and emotional states to some of the shoppers, none of which are included in the code!

More objectively, to what extent do the analytical models, lacking any tracking or analysis of customer movement, reflect the simulations, which model spatial dynamics in detail, including social distancing? This is basically a test of the fundamental assumption of the models that pairwise interactions causing slowdown will occur at the rate of n2.

The most salient aspect of the models is their bifurcated character. This is strikingly similar with respect to the number of customers in the store and the number having completed simulations in Figure 2 and Figure 3 vs. Figure 17 and Figure 18. Both theory and simulation have the same value of Δ when M,c and $A_{1}$ are the same. In the example, if Δ≥41.8 s, there is an equilibrium state, but if Δ<41.8, there is no equilibrium. Moreover, in comparing simulation and theory, the number of customers at equilibrium for 60 s is around 20 for both approaches, and it is around 40 for 42 s.

The average shopping time for each customer having completed shopping decreases when the entry interval increases. Clearly, with fewer customers in the store, shopping is completed more quickly.

In both approaches, the number of customers who complete shopping will increase first and decrease later as the entry interval Δ increases. And the largest number will occur at the bifurcation point.

The conclusion that emerges is that the “n2-interactions” assumption suffices to produce the same bifurcated response to Δ as the more realistic simulation where the interactions are the result only of the random placement of shopping points together with social distancing rules.

Practically speaking, the consequence of this work is to provide a lower limit to the timing of customer entries. The familiar pandemic-era sight of gate-keepers allowing one-at-a-time entry to shoppers in a long queue into the store at the same rate as previous shoppers leave maintains the number of shoppers at a fixed level. But what should this level be? In practice, this has been fixed in terms of limits on the number of shoppers per square metre or by fire safety regulations, or purely subjectively. There have been no previous theoretical considerations, however, of the revenue implications of particular policies in terms of shopping behaviour changes. Our investigation, based on a minimal set of assumptions, suggests that an entry interval resulting in an extension of the average shopping time by a factor of two assures an optimum amount of overall spending. This value is the lowest that will permit a stable “Little’s law” flow of customers. Pertinent empirical or experimental studies that could test this hypothesis are lacking as yet, although our simulations largely confirm our analysis. Our results also shed light on the time course of the approach to equilibrium and of the behaviour of the model when there are somewhat more or fewer shoppers than optimal.

While the concepts incorporated in the models and the simulations have been illustrated with particular values of the parameters, the mathematics remain valid for all positive *M*, *c* and $A_{1}$. There is not likely to be any closed-form solution for $J_{1}$ and the other key quantities in our analysis. A hint of this lies in the connection with the digamma function.

## Figures and Tables

**Figure 1 entropy-25-01668-f001:**
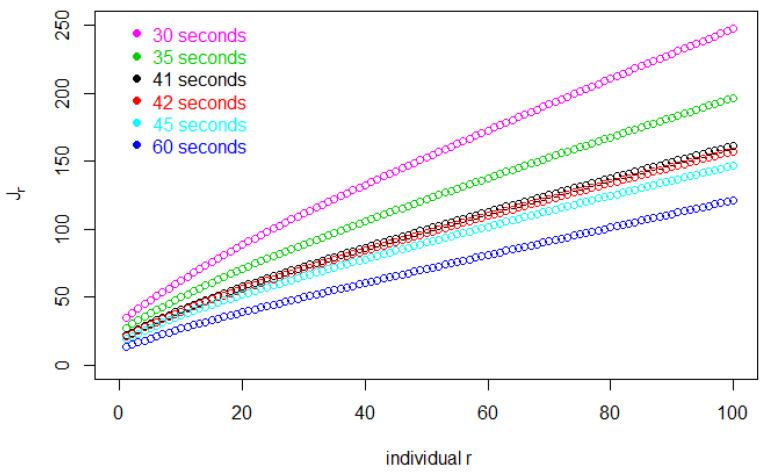
The value of $J_{r}$, the total number of customers entering before individual *r* leaves the store, for the first 100 customers.

**Figure 2 entropy-25-01668-f002:**
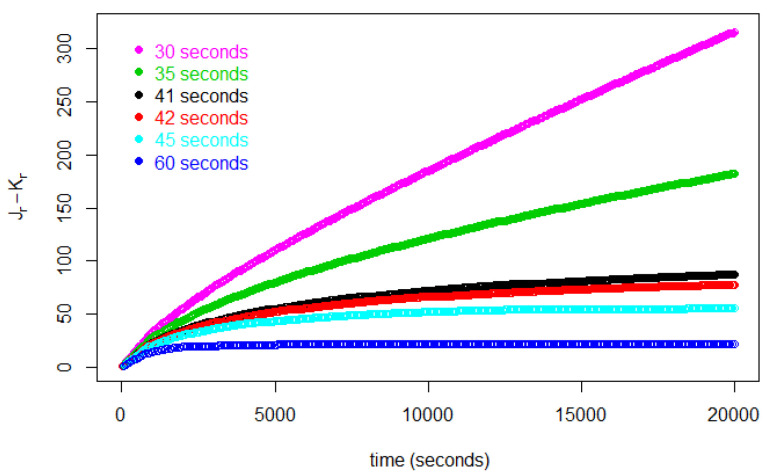
Number of customers in the store during 20,000 s, contrasting equilibrium patterns, Δ≥42 s, with unlimited increase patterns Δ≤41 s.

**Figure 3 entropy-25-01668-f003:**
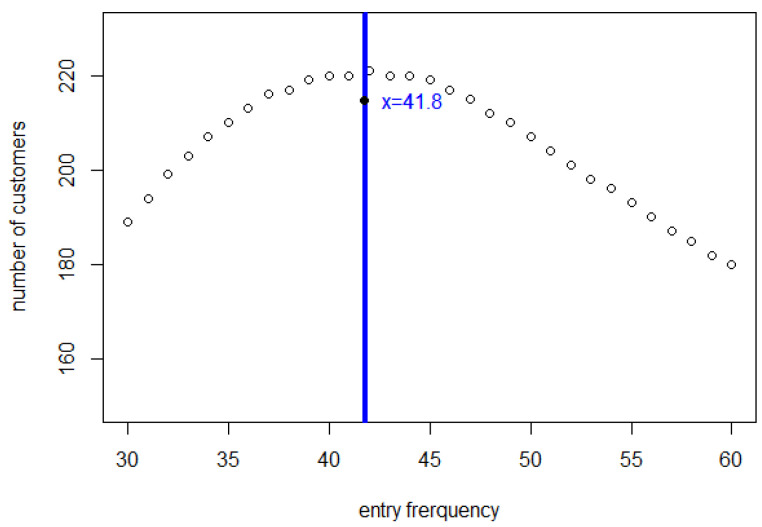
Number of customers who have completed shopping during 5000 s, reflecting the trade-off between increased entry rates and increased shopping times.

**Figure 4 entropy-25-01668-f004:**
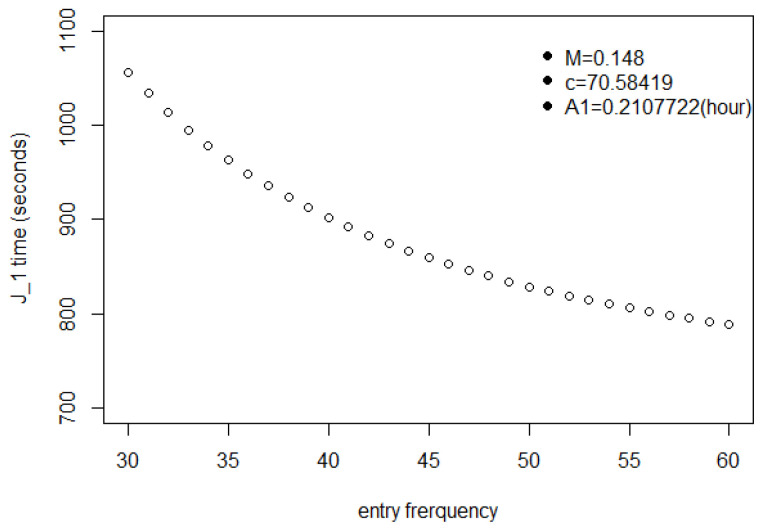
$J_{1}$ as a function of entrance interval, showing substantial slowdown with shorter intervals.

**Figure 5 entropy-25-01668-f005:**
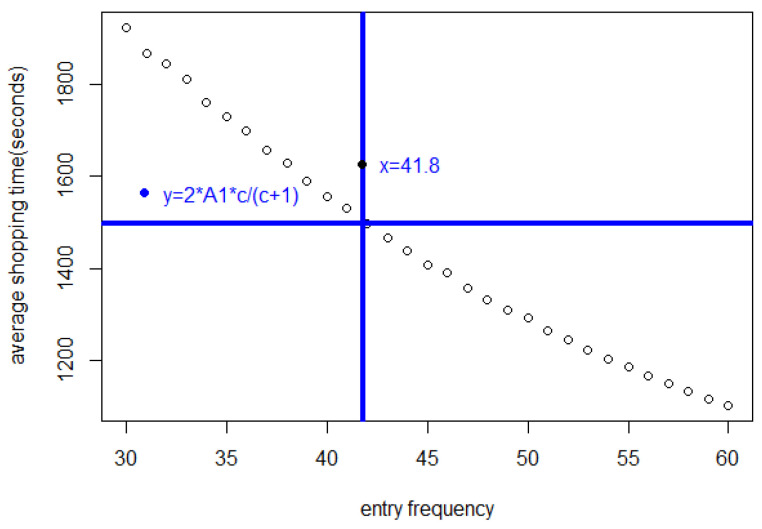
Average shopping time *A* of customers who have completed shopping for different entry intervals, showing that at the predicted optimum Δ=41.8 s, the shopping time nears $A=2A_{1}$.

**Figure 6 entropy-25-01668-f006:**
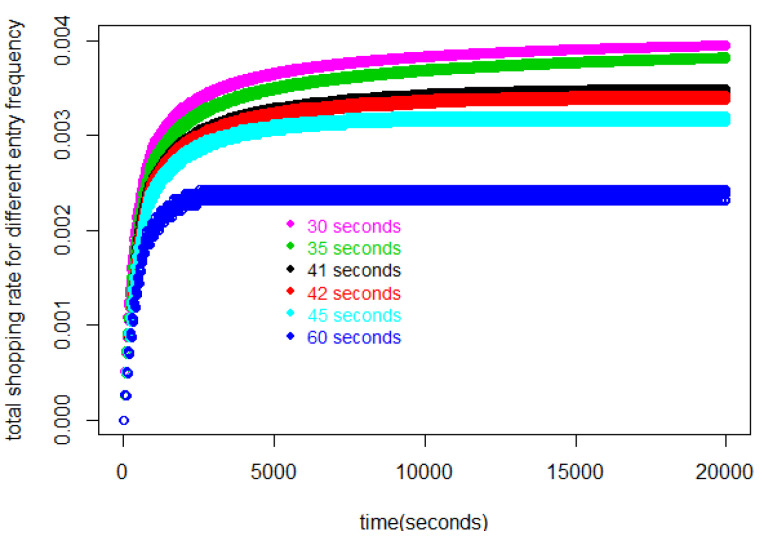
Total spending rate for all customers in the store at each time for various entry intervals.

**Figure 7 entropy-25-01668-f007:**
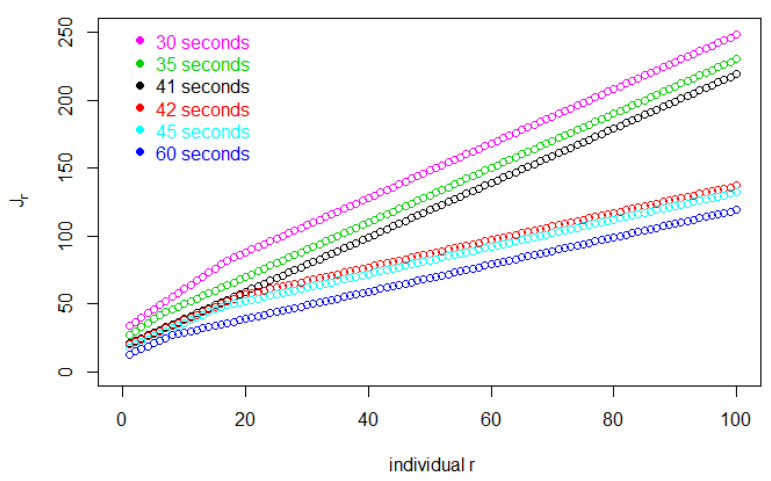
The value of $J_{r}$, the total number of customers entering before individual r leaves the store, for the first 100 customers, in the discrete model.

**Figure 8 entropy-25-01668-f008:**
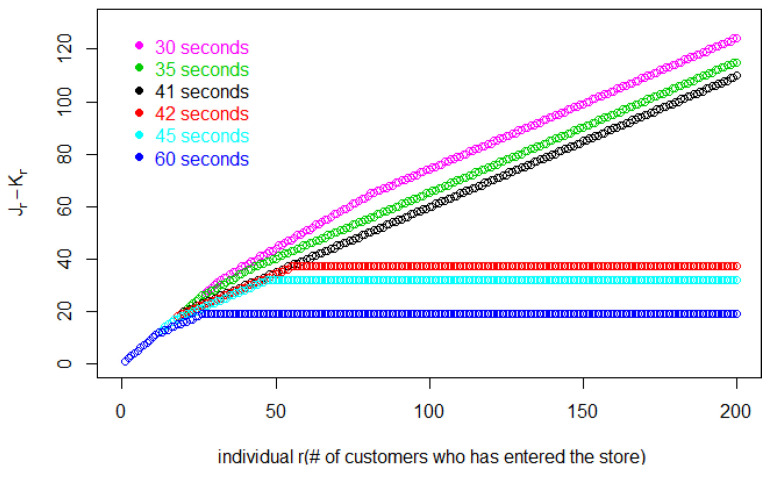
The number of customers in the store when individual *r* enters.

**Figure 9 entropy-25-01668-f009:**
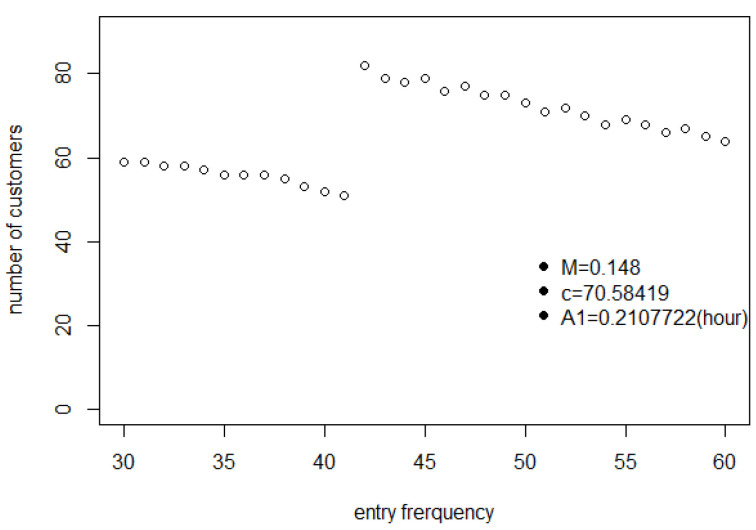
Number of customers who have completed shopping within 5000 s, showing discontinuity at Δ=41 s because of the rapid customer increase and consequential slowdown of shopping.

**Figure 10 entropy-25-01668-f010:**
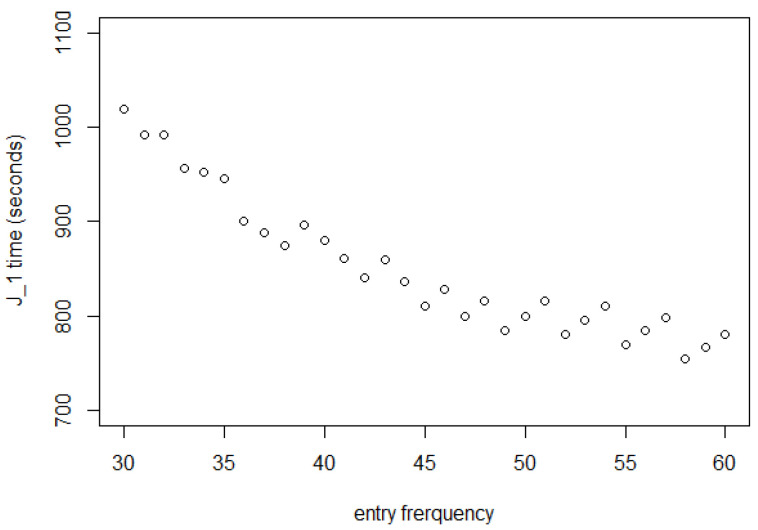
First customer exit time $J_{1}$.

**Figure 11 entropy-25-01668-f011:**
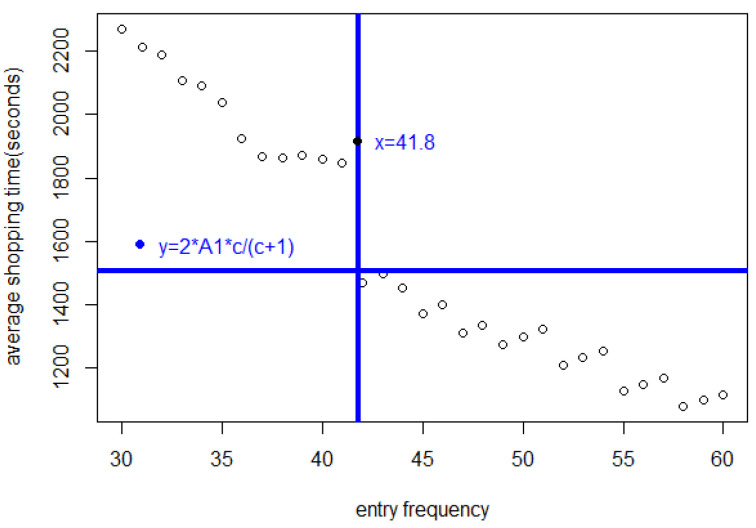
The average shopping time *A* of customers who have completed shopping, for different entry frequencies.

**Figure 12 entropy-25-01668-f012:**
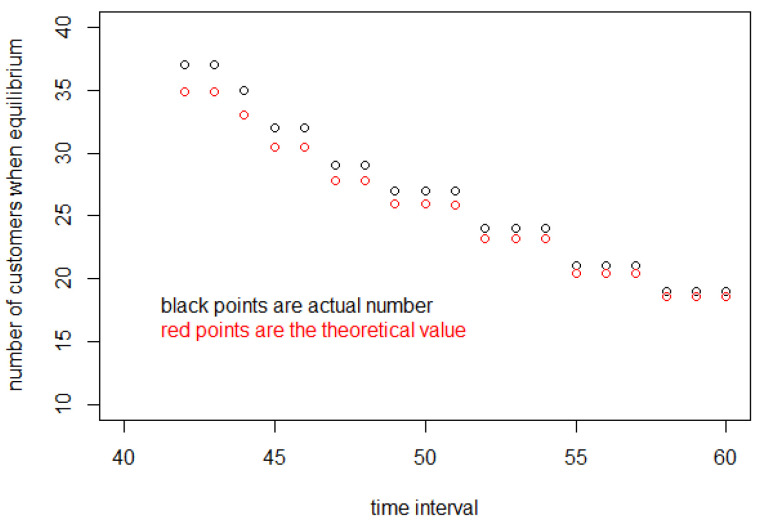
The number of customers in the store when it meets equilibrium.

**Figure 13 entropy-25-01668-f013:**
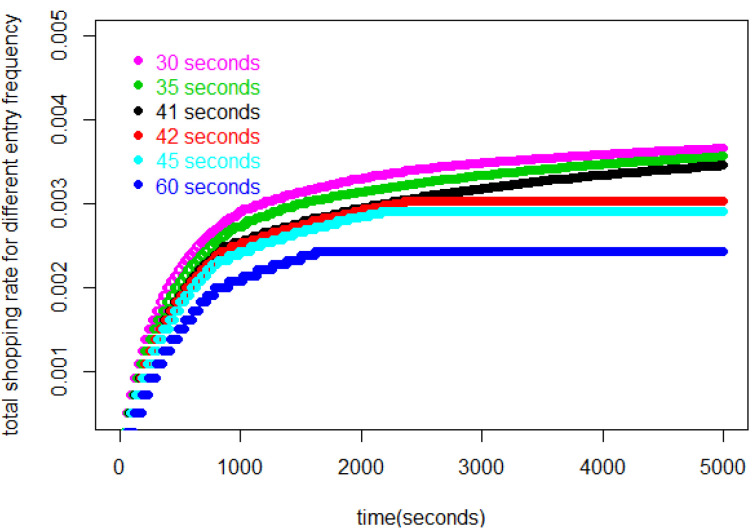
Total spending rate for the customers in the store for various entry intervals.

**Figure 14 entropy-25-01668-f014:**
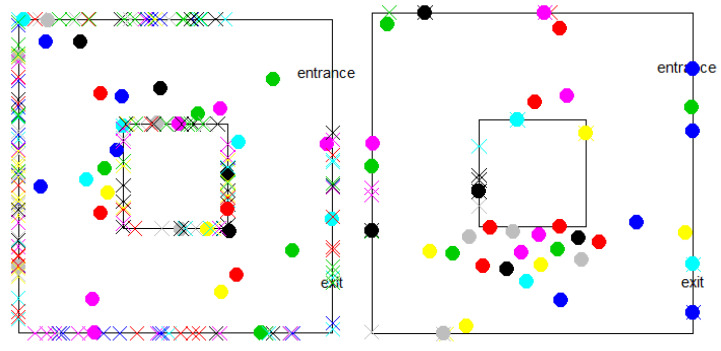
Frames from the animation showing the distribution of customers at shopping points and in the aisles. On the right is an example of freezing. There are 14 customers who cannot move. Note that there are few shopping points left.

**Figure 15 entropy-25-01668-f015:**
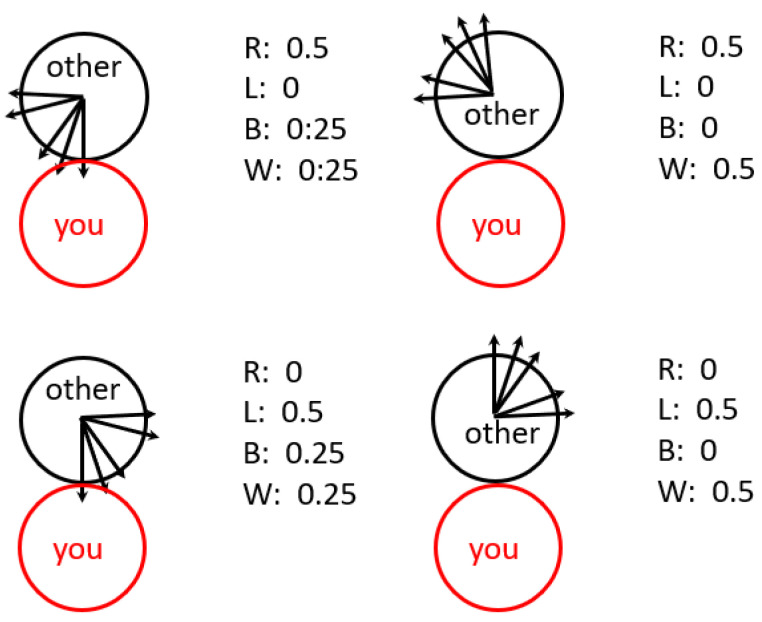
Rules for customers avoiding space infringement. The axis between the two shoppers at the projected point of infringement defines four quadrants in which the “other” person is known to be moving. The probability that “you” chooses right (R), left (L), back (B) or wait (W) depends on the quadrant.

**Figure 16 entropy-25-01668-f016:**
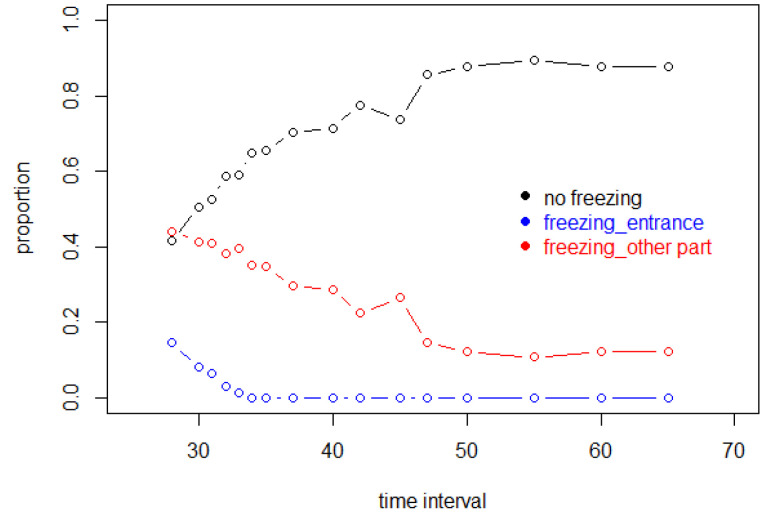
The rate of freezing as a function of entrance interval Δ.

**Figure 17 entropy-25-01668-f017:**
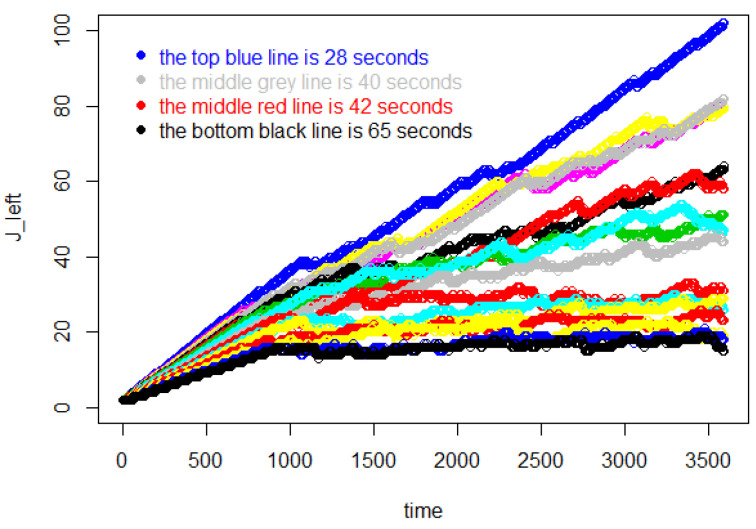
The number of customers in the store during 3600 s.

**Figure 18 entropy-25-01668-f018:**
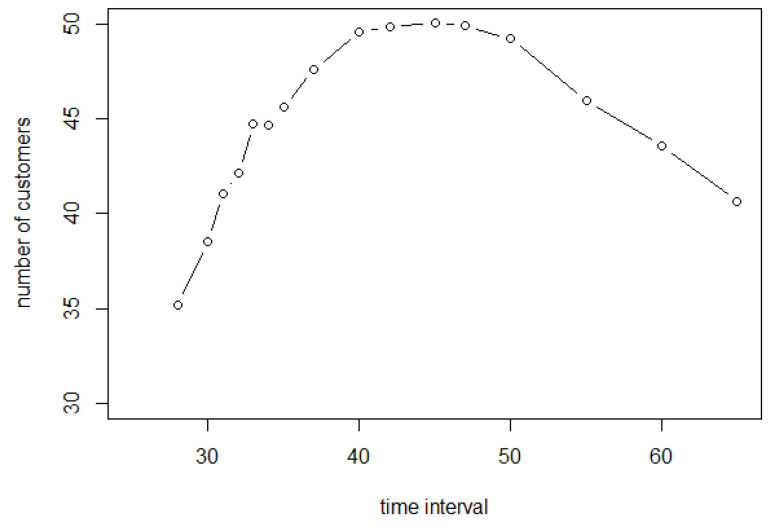
Number of customers who have completed shopping during 3600 s.

**Figure 19 entropy-25-01668-f019:**
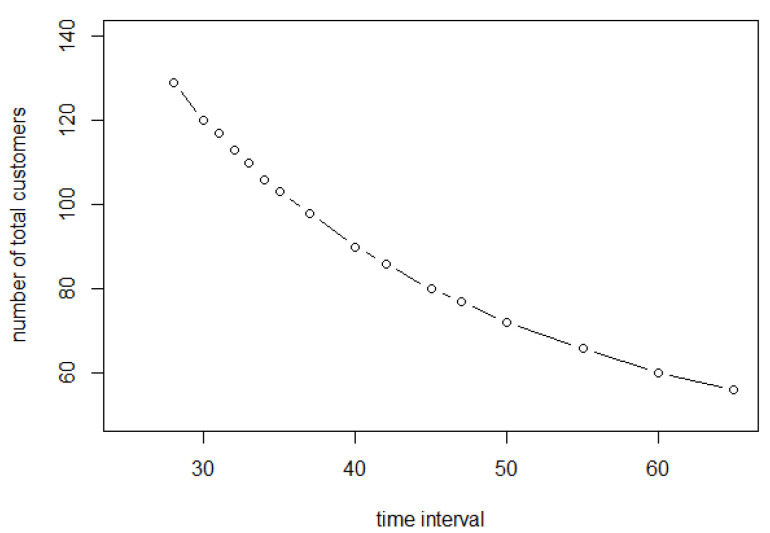
The number of customers who have entered the store within 3600 s.

**Figure 20 entropy-25-01668-f020:**
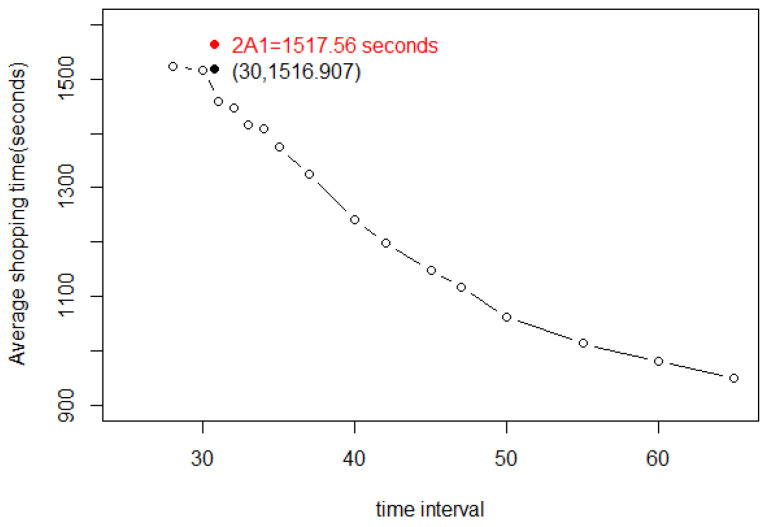
Average shopping time of customers who have completed shopping for different entry frequencies.

## Data Availability

Data are contained within the article.

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
