# Peer review of "Capacity, Collision Avoidance and Shopping Rate under a Social Distancing Regime"

_entropy, 2023, doi:10.3390/e25121668_

Round 1
Reviewer 1 Report
Comments and Suggestions for Authors
entropy-2663963-peer-review
As a first point, the authors need to carefully read through the manuscript and run it through spell checker: see, for instance, lines 27, 55, 116, and 214.
In Sec. 1 they describe the condition on the flow rate for the equilibrium to hold. I personally would not call it a “Theorem,” given that it is a result of simple quadratic equation. More importantly, relations such as eqs. (9) and (10) and in 1.2 should have already been pointed out in 1.1 and the latter should be given in a numbered equation.
Line 121 is incomplete: I am assuming that it should be “To maximize … one needs to maximize flow rate
.” It also should not be split off into a separate paragraph but combined with lines 119 and 120.
In line 122 eqs. For and
should not be repeated inline but just referred to as eqs. (6) and (7). More importantly, the statement in lines 123 and 124 requires clarification. Namely, since realistically
, maximum
is achieved when
In Sec. 2 and 3 the authors describe continuous and discrete models, with similar outcomes. I am assuming that “Theorems 2 and 2” refer to eqs. (25) and (26). If so, this should be explicitly stated and eq. (27) should not be separately numbered and if it is, the preceding equation should be numbered as well.
I am concerned with Figs. 2 and 6. Authors explanation of divergence in Fig. 2 in lines 183-185 is unsatisfactory, given also that spending rate in Fig. 6 displays saturation regardless of whether customer number saturates. This needs to be addressed carefully because otherwise it appears that to maximize the spending rate the store needs to pack customers like sardines in a can.
My understanding is that simulation in Sec. 4 yields similar results to 2 and 3 as per Fig. 19. The authors need to give more detail to interplay of results of Secs. 2-4.
Finally, the comment in lines 583-586 is amusing. Practically, if due to restrictions a store limits the number of customers, they just calculate the number of people allowed in the store and let as many people in as come out regardless of the time interval. If there is no restrictions, there is nothing that the store can do, and customers self-regulate. So, in the end, I am not sure what the authors are trying to say – they should be more precise in their conclusions.
In summary, the manuscript needs revisions to be accepted for publications.

Some editing required, as indicated in review
Author Response
uploaded

Reviewer 2 Report
Comments and Suggestions for Authors
Dear Editor,
I have reviewed the manuscript titled "Capacity, Collision Avoidance and Shopping Rate under a Social Distancing Regime" by Zhong and Sankoff. Here is my feedback for the authors' consideration.
Introduction:
The introduction of your paper provides a comprehensive overview of the problem you intend to address and the approach you plan to take. However, it would benefit from improved clarity and a more concise presentation.
Consider breaking down some of the longer sentences to enhance readability.
The introduction could be more reader-friendly by explicitly outlining the paper's structure, thus giving readers a clear roadmap of what to expect.
Rationale for Research:
While you explain the safety measures imposed during the COVID-19 pandemic and the limitations of Little's Law, it would be helpful to explicitly state why studying the impact of customer intake rates on shopping behavior during a pandemic is important.
Clarify the practical implications or insights you hope to derive from this research.
Research Questions:
Explicitly state the research questions or hypotheses your study aims to answer.
Define the specific outcomes or insights you seek to gain from your research.
Clarify the significance and potential contributions of your research.
Explain what policymakers, businesses, or researchers will gain from understanding the relationship between customer intake rates and shopping behavior during a pandemic. Consider your target audience (fellow researchers, policymakers, or a general audience) to determine the appropriate level of detail and technicality in your introduction.
section 1: Model
Equation 3: The explanation for the second term in Equation 3 (n-1)*(A/c) is unclear.
Define the term and provide its dimensions. Additionally, clarify the definition of the effective area parameter "c."
Sections 2 and 3:
1-Figures: Specify whether the figures in sections 2 and 3 are obtained from calculations or simulations.
Explain the rationale behind the chosen numbers (e.g., M = 0.137, c = 70.58, A1 = 0.2108) for a realistic scenario.Improve figure captions by providing detailed descriptions and, if needed, improve the layout of the figures.
2-Alternative Formulation:
Explain why it is necessary to calculate an alternative formulation of the same problem in section 3.
Clarify whether the results are better or preferable in one formulation over the other.
Discontinuity in Figures 9 and 10:
3-Address the discontinuity in figures 9 and 10, especially regarding why "42" separates two regimes.
Explain the significance of the number "42 seconds," which also appears in figures 1 and 3.
Section 4: Simulation Description:
1-The description of the simulation is not entirely clear.
Consider placing the algorithm in an appendix for improved clarity.
2-Subsection (4.3) regarding reducing the quadratic coefficient for collision detection could be made more reader-friendly.
3-Comparison of Simulated and Theoretical Results:
4-It would be beneficial to compare the simulated and theoretical results side by side to help readers understand the differences and the accuracy of the models.
Final Remarks:
1-Comparison with Empirical Data.
2-Consider comparing your results with empirical data to validate your findings.
Relevance to Journal
3-Provide a clear argument for why the manuscript is suitable for the journal "Entropy" rather than marketing-focused journals.
Clarify the relevance of your results, especially if they present surprising or valuable insights.
In its current form, I cannot recommend the publication of the manuscript in "Entropy." I recommend that the authors work on improving the figures, clarifying the text, and providing more data to demonstrate the relevance of their results.
Author Response
uploaded

Round 2
Reviewer 2 Report
Comments and Suggestions for Authors
Now the manuscript is suitable to be accepted for publication by Entropy.